# Convolutional Neural Network-Based Travel Mode Recognition Based on Multiple Smartphone Sensors

Lin Guo [1,2], Jincai Huang [1,*], Wei Ma [3], Longzhi Sun [2], Lianjie Zhou [3], Jianping Pan [3] and Wentao Yang [4]

1 Big Data Institute, Central South University, Changsha 410083, China; guolincsu@csu.edu.cn
2 School of Computer Science and Engineering, Central South University, Changsha 410083, China; sunlongzhi@csu.edu.cn
3 School of Civil Engineering, Chongqing Jiaotong University, Chongqing 400074, China; weima@cqjtu.edu.cn (W.M.); zlj0808@cqjtu.edu.cn (L.Z.); panjianping@cqjtu.edu.cn (J.P.)
4 National-Local Joint Engineering Laboratory of Geospatial Information Technology, Hunan University of Science and Technology, Xiangtan 411100, China; yangwentao8868@126.com
* Correspondence: huangjincaicsu@csu.edu.cn

**Abstract:** Nowadays, large-scale human mobility has led to increasingly severe traffic congestion in cities, how to accurately identify people's travel mode has become particularly important for urban traffic planning and management. However, traditional methods are based on telephone interviews or questionnaires, which makes it difficult to obtain accurate and effective data. Nowadays, numbers of smartphones are equipped with various sensors, including accelerometers, gyroscopes, and GPS, providing a novel social sensing data source to detect people's travel modes. The fusion of multiple sensor data is a promising way for travel mode detection. However, how to use these sensor data to accurately detect travel mode is still a challenging task. In this paper, we presented a light-weight method for travel mode detection based on four types of smartphone sensor data collected from an accelerometer, gyroscope, magnetometer, and barometer, and a prototype application was developed. Then, a novel convolutional neural network (CNN) was designed to identify five representative travel modes (walk, bicycle, bus, car, and metro). We compared the overall performance of the proposed method via different hyperparameters, and the experimental results show that the F value of the proposed method reaches 97%, which verified the effectiveness of the proposed method for travel mode classification.

**Keywords:** traffic management; social sensing; travel mode; CNN; smartphone data

## 1. Introduction

Understanding different travel modes of human travel will benefit numbers of applications such as smart city, urban transportation planning, and urban traffic management. As a part of the concept of smart city, smart mobility aims to manage transportation infrastructure more efficiently and provide more efficient transportation services, which will need the big data to understand how people travel and how the transportation is used [1,2]. For example, understanding and modelling different travel modes will shorten the travel time and help to reduce car traffic congestion on the most loaded routes by introducing collective transport, which will also reduce the air pollution. Moreover, this will also help to track human mobility, which enables the monitoring of the spread of diseases and other potential hazards [2]. Traditional methods to collect travel mode data of human mobility are based on household surveys or telephone interviews for over four decades, which are time-consuming and labor-intensive [1]. Furthermore, the collected data tends to be inefficient and inaccurate if low response and incomplete information are involved. Since the mid-1990s, GPS (Global Positioning System) data has become popular for travel mode detection [1–3], but GPS signals are easily lost in indoor and high-rise urban environments due to the impact of multipath [4]. For example, if the target travel mode is walking

indoors or in a shopping mall, it is difficult for GPS data to support the identification of the "walking" travel mode.

With the rapid development of mobile computation, a smartphone-based method is becoming a promising way for travel mode recognition. Smartphones with built-in sensors can be considered as primary motion capture sensors [5]. Their enormous market penetration rate and being relatively close to users is another advantage for travel mode detection [1]. Recently, a number of smartphone-based activity detection methods have been developed [6–11]. It should be noted that individuals' activity modes can reflect their travel mode. Thus, many studies of smartphone-based approaches on classification of travel modes have been proposed [2,12–14]. A smartphones-based travel modes recognition method generally includes two steps: the first step is activity features computing (e.g., mean, variance, and maximum) based on raw sensors data, and for the second step, the extracted features are adopted to classification algorithms for estimating the travel mode. The travel mode recognition performance depends highly on the feature representation of the sensor data [12–14]. However, current methods rely on complex hand-crafted features extraction, which usually require laborious human intervention. The feature selection process is costly and inefficient. To solve this problem, researchers exploit deep learning algorithms to extract multiple levels of feature automatically [15–18].

In this paper, to develop a convenient prototype application of travel mode recognition and compare the effectiveness of different sensor data fusion for travel mode recognition, we proposed a convolutional neural network (CNN) based method for travel mode recognition by fusing multiple types of smartphone sensor data. Four types of sensors were selected to detect users' travel modes, which contained accelerometer, gyroscope, magnetometer, and barometer. Compared with conventional machine learning techniques based on manual selected features, the proposed method was able to automatically extract different moving features. We also developed a prototype application to collect smartphone sensor data which did not need to set an initial position or orientation value while the application is running. The collected smartphone sensor data was under five different transportation modes (walk, bike, bus, car, and metro). In the experiment analysis part, we compared and analyzed the performance of the proposed system under different hyperparameters and combinations of sensors. The results showed that the proposed approach was able to recognize different travel mode of users more efficiently and accurately.

The major contributions of this paper can be summarized as follows.

(1) We designed a CNN-based travel mode recognition system, which consisted of data collection, data segmentation, and deep learning model. It could provide accurate and energy-efficient transportation mode detection ability based on multiple smartphone sensors.

(2) We evaluated the classification performance of the proposed CNN model under different hyperparameters and combinations of sensors. We also discussed the impact of different sensor combinations on the accuracy of travel pattern recognition. Experimental analysis provided a constructive and helpful reference for sensors data selection and fusion in future research.

The organization of this paper is as follows. The related works are reviewed in Section 2. Section 3 discusses the architecture of the proposed methods. Experimental results and analysis are described in Section 4. The conclusions are then presented in Section 5.

## 2. Related Works

Numbers of studies have proposed methods to infer transportation modes based on multiple data sources. One popular data source is GPS data [19–25], which are real-time records of people's activities of travel [19–21], thus providing a crucial data source for travel mode detection. The GPS data-based methods are popular to classify multiple travel modes based on different moving features. Zheng et al. [22] proposed a framework to infer transportation mode from users' GPS trajectories. In their work, each trajectory was

divided into segments by a change-point-based segmentation algorithm and features of each segment were extracted. Then, a traditional classification algorithm was deployed to train the transportation mode detection model. Maenpaa [23] intended to select the most significant features from three sets of potential features extracted from GPS trajectories. Bolbol et al. [24] proposed a GPS-based transportation mode detection method using a moving window SVM classification. They implemented the proposed method by using a framework based on SVM classification. Xiao et al. [25] detected different travel modes based on GPS track data and Bayesian networks. These studies mainly rely on a single GPS data source, and it is difficult to effectively classify travel modes for areas where the signal is blocked. Especially, when people do not use GPS sensors (such as subway travel, walking), the above methods are difficult to accurately classify travel modes.

Instead of single GPS-based methods, some researchers improved the travel mode classification performance by using multiple sources combination. Gong et al. [26] proposed a GPS/GIS method for travel detection in New York City. Wang et al. [27] developed and evaluated a Random Forest classifier combined with a rule-based method to detect travel modes. Besides, socioeconomic attributes were used to further improve the precision. Feng and Timmermans [28] utilized a Bayesian Belief Network to combine GPS and accelerometer data for travel mode detection.

At the methodological level, deep learning-based methods currently have been proposed to integrate the hand-crafted features with high-level features [29–31] for transportation mode estimation. Kim et al. [31] proposed a multi-feature network for sound classification. Endo et al. [32] utilized a fully connected deep neural network (DNN) to automatically extract high-level features. Similarity, Wang et al. [33] obtained the deep features by transforming point-level features to deep features with the aid of sparse auto-encoder. Dabiri and Heaslip [1] proposed a transportation mode detection method based on GPS trajectories using a convolutional neural network that structures a raw GPS trajectory into a format. The key of these methods is to convert a raw GPS trajectory into a 2D image structure as the input of the DNN model. However, GPS-based methods show poor performance in a complex urban environment where a significant multipath effect exists, and the problems mentioned above are still difficult to solve [26].

With the development of mobile computation, various sensors carried in smart phones are becoming novel data sources for crowd behavior identifying and environment perception [34–37]. Thus, smartphone-based methods have gradually emerged as a popular way to categorize different travel modes [11]. Reddy et al. [12] presented a convenient classification system that utilized a smartphone with a built-in GPS receiver and an accelerometer. The proposed system consisted of a decision tree followed by a first-order discrete Hidden Markov Model. Hemminki et al. [2] proposed an accelerometer-based transportation mode detection using smartphones. Their paper described a novel gravity estimation technique for accelerometer measurements to distinguish information pertaining to movement behavior from other factors that affect the accelerometer signals. They also introduced a new set of accelerometer features for improving the detection performance of transportation mode detection. Zhou et al. [14] developed and evaluated a chained random forest mode that classifies smartphone data into different travel modes without prior segmentation. Fang et al. [4] utilized deep neural network to identify the transportation modes of smartphone users, whereas, the training process is very time consuming. Shin et al. [13] presented a smartphone-based application for urban data collection, sensing and classifying of transportation mode. Assemi et al. [35] developed a statistical mode integrated with smartphone travel surveys to identify travel modes. Su et al. [36] proposed a GPS and network-free method to detect travel mode using smartphone sensors. The methods mentioned above provide efficient time and frequency domain features for travel pattern recognition.

To sum up, the current research based on smartphones mainly uses the behavioral characteristics shown in sensor data to classify travel modes, which is difficult to solve the combination problem of multi-source sensor data, and the effectiveness of multi-source

sensor data combination is unclear. Therefore, in this paper, a travel mode classification method based on CNN model was proposed, and based on the combination between different sensors, the hyperparameter analysis and effectiveness evaluation of the CNN model were carried out. This study will provide exploratory research examples for the application of ubiquitous smartphone sensors in the field of smart transportation.

## 3. Methods

### 3.1. Architecture

To construct a prototype system to detect users' travel modes, we developed an Android application that automatically collected user's moving data by smartphones. The architecture of the system is demonstrated in Figure 1.

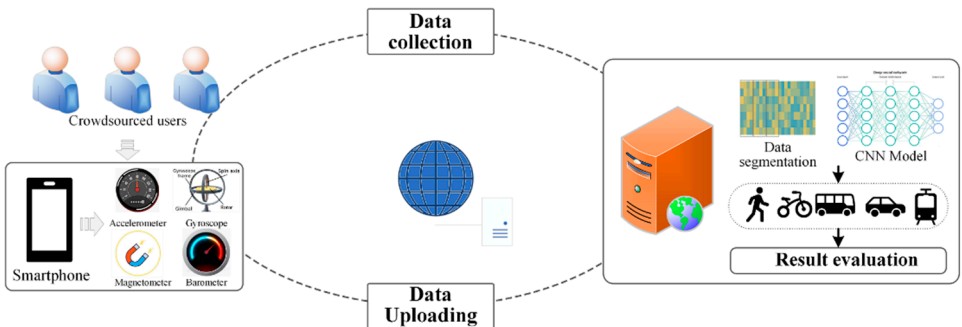

**Figure 1.** The architecture of the proposed system.

As shown in Figure 1, the travel mode detection system mainly consists of two parts: the mobile terminal and server terminal.

(1)    Mobile terminal

We developed an Android application to collect four different sensors data, which includes accelerometer data, gyroscope data, magnetometer data, and barometer data at the mobile terminal. Through the mobile phone Android application, users can collect and upload their own mobile phone sensor data.

(2)    Server terminal

The web server can monitor and query the sensor data uploaded by the user in real time, and data processing algorithms are built into the server, including data segmentation and CNN models. Firstly, the time series data are separated according to a specific time window before being used to construct a detection model. Then, the datasets are used as the input of convolutional neural network (CNN) for travel mode (walk, bike, bus, car, metro) recognition.

### 3.2. Data Collection

We developed an Android application to collect the data related to user movement. Figure 2 shows a portion of the data collected by accelerometer, gyroscope, magnetometer, and barometer.

Based on the developed Android application, we collected the datasets by smartphone sensors under five representative travel modes (walk, bike, bus, car, and metro). Users were able to record different modes of data by choosing the corresponding transportation mode. The data sampling frequency was set 50 Hz. Moreover, the phone did not require us to set an initial position or orientation while the application is running.

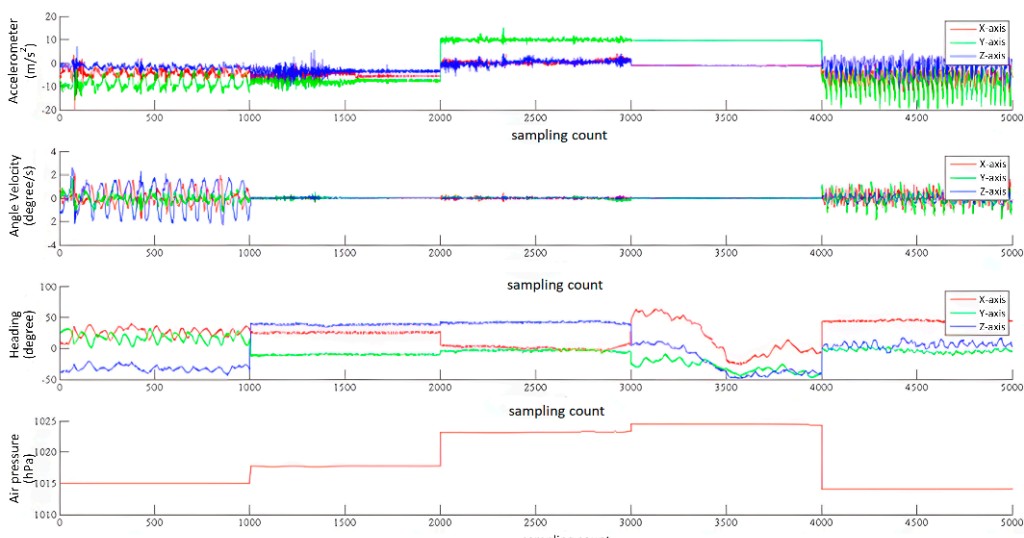

**Figure 2.** The data collected by accelerometer, gyroscope, magnetometer, and barometer.

### 3.3. Data Segmentation

The raw sensor data collected from smartphone sensors are a typical type of time series data. We first partitioned the sensor data into segments according to a certain time window. The accelerometer data contained three coordinate dimensions of x, y, and z, so did gyroscope, and magnetometer, whereas the barometer had only one dimension. From the perspective of time series, the data generated by a user under a certain travel mode can be expressed as:

$$d_t = \left( a_t^x, a_t^y, a_t^z, g_t^x, g_t^y, g_t^z, m_t^x, m_t^y, m_t^z, b_t \right)$$

$$d_{t+1} = \left( a_{t+1}^x, a_{t+1}^y, a_{t+1}^z, g_{t+1}^x, g_{t+1}^y, g_{t+1}^z, m_{t+1}^x, m_{t+1}^y, m_{t+1}^z, b_{t+1} \right)$$

$$d_{t+N} = \left( a_{t+N}^x, a_{t+N}^y, a_{t+N}^z, g_{t+N}^x, g_{t+N}^y, g_{t+N}^z, m_{t+N}^x, m_{t+N}^y, m_{t+N}^z, b_{t+N} \right)$$

where $(a_t^x, a_t^y, a_t^z)$, $(g_t^x, g_t^y, g_t^z)$, and $(m_t^x, m_t^y, m_t^z)$ represent the data collected by accelerometer, gyroscope, and magnetometer, respectively. $b_t$ represents the data collected by barometer. $N$ is the size of the data sample. We grouped the sensors data into a window of samples with a size of 2 s, thus, 10 features are extracted for each time stamp. Figure 3 shows the heatmap of the time series data of 10 features.

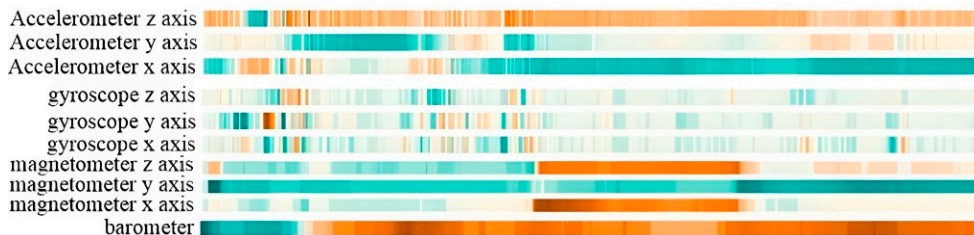

**Figure 3.** The time series data collected by smartphones.

### 3.4. Travel Mode Detection Model

The workflow of the proposed method for travel model recognition is presented in Figure 4. The collected data was firstly preprocessed, then all input data was trained by a convolutional layer and pooling layer, followed by a full connected layer. Finally, the SoftMax layer was designed for classifying different travel modes.

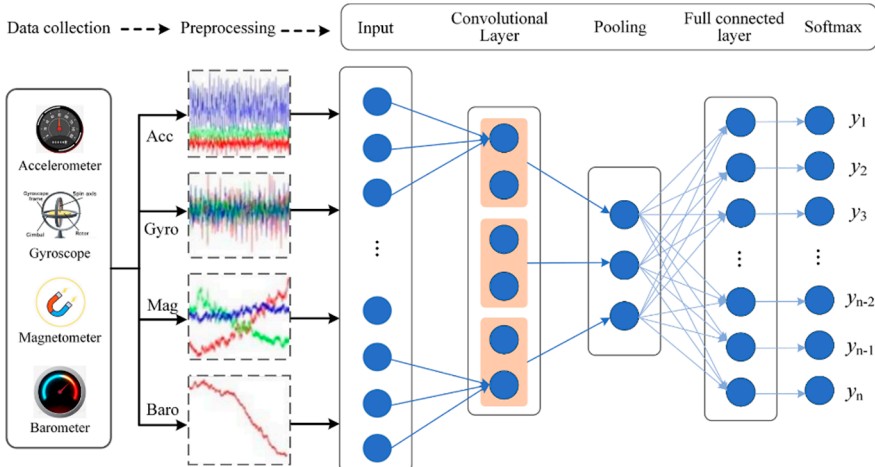

**Figure 4.** The workflow of the proposed CNN-based method.

### 3.4.1. Preprocessing

To avoid the vanishing gradient problem, we normalized the input data to improve the optimization and accelerate convergence. For the non-uniform distribution of input data in each dimension, the normalization can solve the problem by changing the distribution of the input data. If an sample was $\hat{x} = [x_1, x_2, \ldots x_h]$, the mean is $\mu$ and the variance is $\sigma^2$, where $\mu = \frac{1}{m} \sum_{i=1}^{m} x_i$, $\sigma^2 = \frac{1}{m} \sum_{i=1}^{m} (x_i - \mu)^2$.

we normalize the sample data by:

$$\hat{x} = \frac{x - \mu}{\sqrt{\sigma^2 + \varepsilon}}$$

where $\varepsilon$ is an infinitely small value.

### 3.4.2. CNN Model

As illustrated in Figure 5, the CNN is designed for one dimensional sensor data, which combines four types of sensor data (total 10 features of accelerometer, gyroscope, magnetometer, and barometer). The proposed CNN consists of three types of layers: convolutional layer, pooling layer, and fully connected layer, as shown in Figure 5.

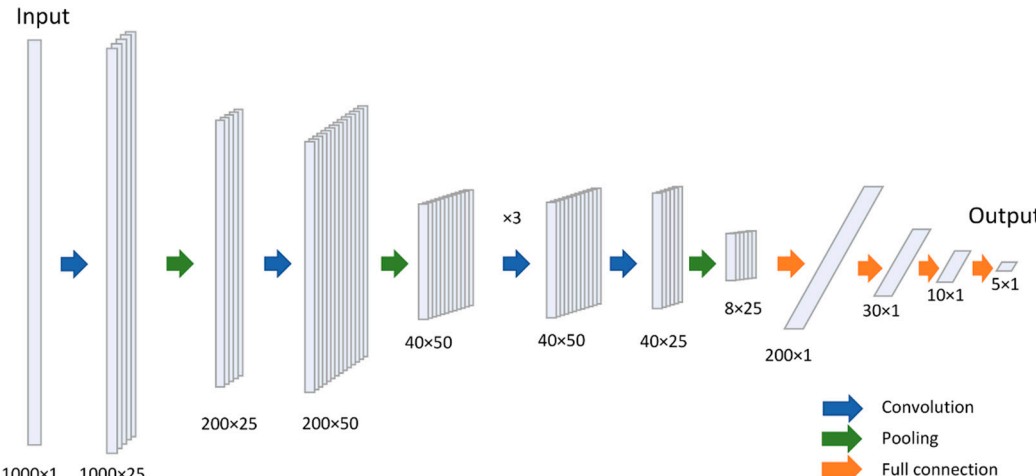

**Figure 5.** The architecture of CNN.

For convolutional layer, the input data is a $1000 \times 1$ vector, which contains four types of sensors data. The input vector can be described as:

$$x = [x_0, x_1, \ldots x_{n-1}]$$

where $n$ is the number of data in a sliding window ($n = 1000$). Then, we perform a convolution operation on the input data. The filter $w$ and the output of the convolutional layer $z$ are:

$$w = [w_0, w_1, \ldots w_{m-1}]$$

$$z = [z_1, z_2, \ldots z_h]$$

where $m$ is the filter size, and the $h$ is the length of the output data. For the $j$th element $z_j$, there is:

$$z_j = \sum_{i-1}^{m-1} w_i x_{i+b*j}$$

where $b$ refers to be the stride length of convolution. The relation between $h$, $m$, $n$, and $b$ can be expressed as:

$$h = \left(\frac{n-m}{b}\right) + 1$$

Afterward, an extra activation layer is added to improve the express capability of the network, the output of activation layer is $a = f(z)$, where $f$ is the activation function. We use Relu as the activation function.

For the pooling layer, we choose the statistic nearby inputs value to extract more robust features. The maximum, minimum or mean value of a given region is computed as the output of pooling layer for reducing the length of data, simplifying computing, and avoiding over fitting. Max pooling is utilized in the proposed network.

The fully connected layer is followed as the classifier to recognize different travel modes. The softmax layer is applied to map the result of fully connected layer in the range of [0, 1], which can be thought of as the probability that the input data belongs to a certain type of travel mode. The output of fully connected layer is $f = [f_1, f_2, \ldots f_s]$, and the output of the softmax is $y' = [y'_1, y'_2, \ldots y'_s]$, which indicates the probability of travel mode:

$$y'_i = \frac{e^{f_i}}{\sum_{k=1}^{s} e^{f_k}}$$

We adjust the weights and biases in each layer by minimizing the loss between the initial prediction and the label to get the best prediction model, the loss can be described as follows:

$$h_{y'}(y) = -\sum_i y'_i \log(y_i)$$

where $y$ is the label of sample and $y'$ is the output of the network. In this study, we applied the Adam optimization algorithm to solve this least square problem. For training the sensors data, we chose Mini-batch Gradient Descent to optimize the collected data and we set the power of 2 as the batch size.

### 3.4.3. Evaluations

To evaluate the performance of the proposed model, we use the F-measure to evaluate the result, which is defined:

$$F = \frac{2 * P * R}{P + R}$$

where $p$ refers to the precision and $R$ is the recall. $p$ represents the percentage of samples correctly identified as the corresponding travel mode within all samples, and $R$ is the accurately predicted percentage of samples in all labeled samples.

Table 1 listed all the candidate hyperparameters in the proposed method. During the experiments, we set the hidden layers share the same hyperparameters. We evaluated the influence of different hyperparameter sizes on the CNN by adjusting the filter size, number of feature maps, pooling size, learning rate and the batch size. Finally, we chose the best parameter setting as the configuration.

**Table 1.** The hyperparameters of the proposed CNN model.

| Hyperparameters | Range | Values |
|---|---|---|
| Number of convolutional layers | [2, 5] | 2, 3, 4, 5 |
| Filter size | [2, 6] | 2, 3, 4, 5, 6 |
| Number of feature maps | [40, 80] | 40, 50, 60, 70, 80 |
| Pooling size | [2, 6] | 2, 3, 4, 5, 6 |
| Learning rate | [0.0001, 0.005] | 0.0001, 0.0005, 0.001, 0.005 |
| Batch size | $[2^4, 2^8]$ | 16, 32, 64, 128, 256 |

## 4. Results

### 4.1. Dataset and Experimental Setup

The datasets were collected by several Android-based mobile phones, five participants were invited to collect the sensor data in Shenzhen. They collect data separately under different travel mode through the developed Android application.

The sampling rate of the collected data is 50 Hz. The size of the collected data is shown in Table 2. In the experiments, 60% of the processed samples are used to train the CNN model. The proportion of the validation data set is 20%, which is used to select the optimal hyperparameters, and the last 20% of the data is set as the test data to verify the effectiveness of the model in the real world. We did not compare with benchmark data, because the sensors used in the recent research are inconsistent with ours in number and type.

**Table 2.** Datasets.

| Travel Mode | Bike | Bus | Car | Metro | Walk |
|---|---|---|---|---|---|
| Data size | 5572 | 5560 | 5567 | 5367 | 5392 |

### 4.2. Hyperparameters Analysis

We analyzed the effect of the model under six different hyperparameters. Figure 6 shows the F-measure in each activity by using different hyperparameters. The *x*-axis of Figure 6 is F-measure, and the *x*-axis is the travel models.

Experimental results show that the model with three convolutional layers outperforms other results in each type of travel modes in Figure 6a. The experimental results show that when the number of convolutional layers in the CNN model is less than 3 layers, it is difficult to effectively extract features for classification, and when the number of convolutional layers is more than 3 layers, it is prone to over-fitting problems. Therefore, the results show that the network structure of the three convolutional layers can better balance the over-fitting phenomenon and the classification accuracy.

We set the number of feature maps from 40 to 80 and compared the effectiveness of the travel mode classification results. Figure 6b shows the influence of different feature map numbers on the model classification. Feature map is the key to extracting features from data. It is not the more the better, nor the less the better. It is noted that the best performance for each travel state is achieved when the feature map is set as 70.

We performed an experimental analysis of different filter sizes. Figure 6c shows the classification performance based on different filter size. We tried the filter size as 2, 3, 4, 5, and 6, respectively. The experimental results show that the travel mode classification results are better when the filter size is 2, 4, 6, and the best performance was obtained when

the filter size is 2. We chose 3, 4, 5, 6, and 7 as pooling size in our experiments. Figure 6d demonstrates the classification performance for each travel mode with different pooling sizes. According to Figure 6d, the classification achieves the best performance when the pooling size is set to 3. We finally set the pooling size as 3.

Figure 6e demonstrates the classification performance for each travel mode with different batch size. As we known, the batch size is commonly set to $2^n$. We choose candidate of batch size from $2^4$ to $2^8$. It can be seen from Figure 6e that the performance is obviously inferior to other values when the batch size is 128 or 256. In fact, the performance was improved by the increasing of the batch size at the first stage. The classification achieved the best performance when the batch size increased to 64. Then, the performance was decreased by the increasing of the batch size. Therefore, we set the batch size as 64.

Figure 6f show the performance of the proposed model of detecting each types of travel mode based on different learning rate. The algorithm shows poorer performance when the learning rate is less than 0.0005 or bigger than 0.001.

The reason for the bad performance of big learning rate is that the variables update too fast to change to proper gradient descent direction timely. Given more iterations, the tiny learning rate can achieve better performance. However, tiny learning rate is also not the best choice because it means the slow update of the variables and it leads to time consuming of the training. At the end, we set the learning rate is as 0.001.

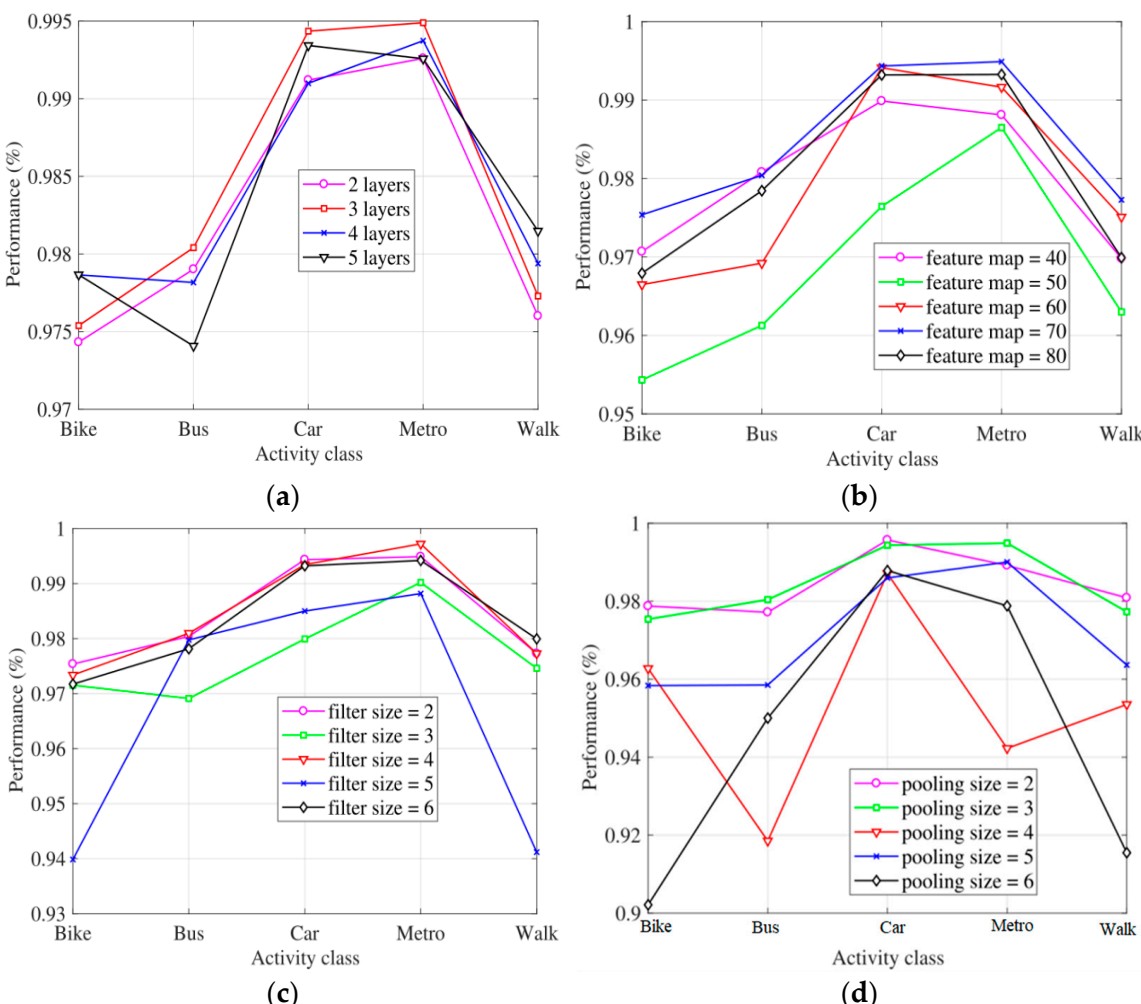

**Figure 6.** *Cont.*

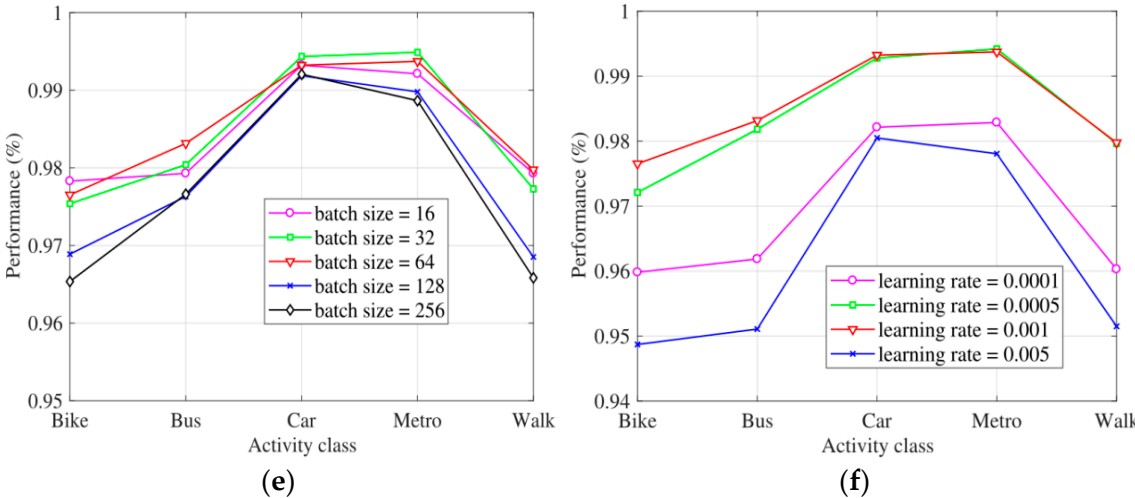

**Figure 6.** The performance under different hyperparameters (**a**): number of convolutional layers; (**b**): number of feature maps; (**c**): filter size; (**d**): pooling size; (**e**): batch size; and (**f**): learning rate.

### 4.3. The Impact of Different Combinations of Sensors

It is an interesting and instructive study on which sensors and how to combine the sensors to effectively identify the user's travel modes. At present, there have been studies using a variety of sensors for travel mode recognition [26–28]; however, these studies fail to answer the question of the effectiveness of sensor combinations. Therefore, the combinatorial effectiveness between the adopted sensors is investigated in this paper.

In the experimental analysis, we evaluated the performance under different combinations of the sensors to discuss the impact on travel mode detection. We combine the four sensors used in this paper from a quantity of 1 to a quantity of 4. The results are shown in Table 3.

From the results in Table 3, we can know that the more types of sensors combined, the better the result. Specifically, combining the accelerometer, gyroscope, magnetometer, and barometer readings achieves the best accuracy (98.5% measured by F-measure). The combination of the accelerometer, gyroscope and magnetometer outperforms other combinations of three types of sensors. For combinations of two types of sensors, the combination of accelerometer and gyroscope achieves the best accuracy. When considering the individual sensors used, the accelerometer achieves a better recognition performance than the other sensors.

**Table 3.** Comparisons of different sensors.

| Number of Sensors | Sensors | F-Measure |
|:---:|:---:|:---:|
| 1 | Acc. | **0.978** |
| | Gyro. | 0.914 |
| | Mag. | 0.852 |
| | Baro. | Nan |
| 2 | Acc. + Gyro. | **0.984** |
| | Acc. + Mag. | 0.983 |
| | Acc. + Baro. | 0.973 |
| | Gyro. + Mag. | 0.916 |
| | Gyro. + Baro. | 0.812 |
| | Mag. + Baro. | 0.867 |
| 3 | Acc. + Gyro. + Mag. | **0.984** |
| | Acc. + Mag. + Baro. | 0.983 |
| | Acc. + Gyro. + Baro. | 0.979 |
| | Gyro. + Mag. + Baro. | 0.915 |
| 4 | Acc. + Gyro. + Mag. + Baro. | **0.985** |

We also discussed the impact of different sensors fusion on accuracy under each type of travel mode. The result was shown in Figure 7. It can be seen that the combination of the accelerometer and the gyroscope can significantly improve the classification performance of Bike, Bus, Car and Walk compared to the results using only the accelerometer. The performance of Bus and Car were further improved after the magnetometer was considered. The barometer increased the recognition precision of Bus.

For several travel modes, the more sensors combination did not work better. Such as Car, the three combinations of accelerometer, gyroscope, and magnetometer achieved the best performance, conversely, the combination of four sensors reduces the accuracy of travel mode detection. In most cases, the combination of four sensors can achieve better performance.

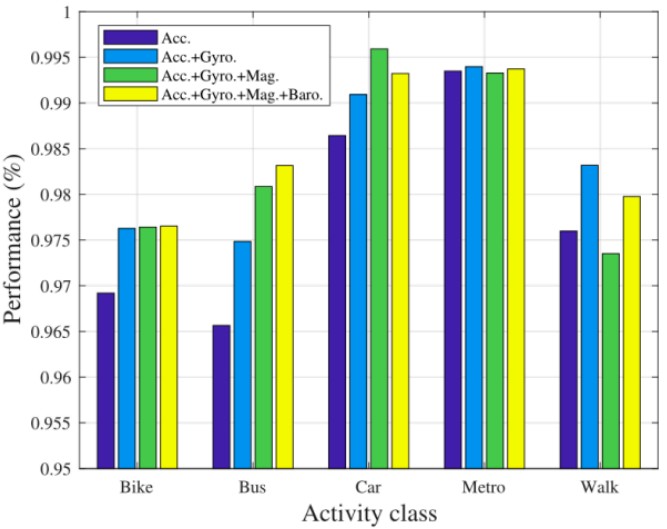

**Figure 7.** The performance of different travel modes with different sensor combinations.

### 4.4. Result of Travel Mode Detection

Based on the parametric analysis, we showed the result of travel mode detection under the optimal hyperparameters on the test data. Figure 8 shows the performance of our method in recognizing each type of travel mode.

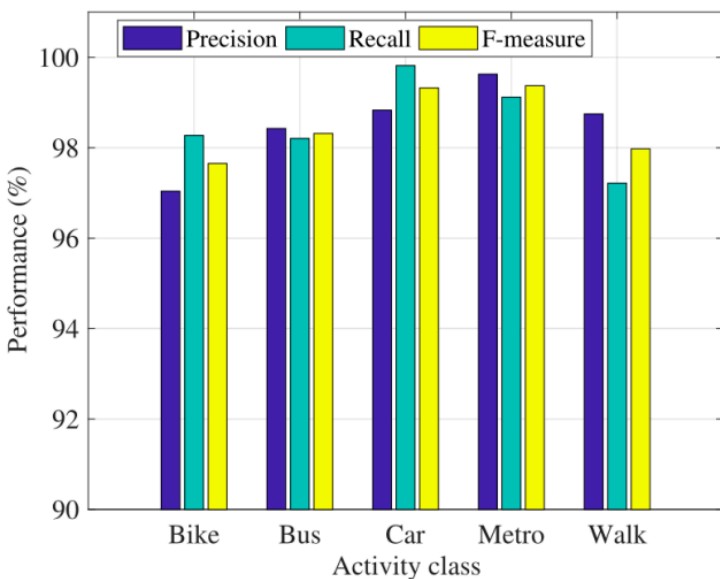

**Figure 8.** The performance of the proposed method.

From Figure 8, we can see that the proposed method shows excellent performance in recognizing all travel modes. It can be found that the F value of all five types of travel mode recognition is above 97%. The same goes for precision and recall. The effect of travel mode "car" is the best, with precision, recall rate and F value reaching more than 98.5%. Walking mode has the lowest accuracy among the 5 types of travel modes, which may be caused by the randomness and confusion of walking behavior. The F-measure values for all the activities are equal or higher than 0.98 except bike, whose F-measure is 0.97.

Table 4 shows the confusion matrix of the proposed method on the test data with the optimal hyperparameters. Table 4 shows that the algorithm can recognize these five types of travel mode with few confusions between different types of travel modes.

**Table 4.** The confusion matrix of the proposed method.

| Predicted<br>Actual | Bike | Bus | Car | Metro | Walk |
|---|---|---|---|---|---|
| Bike | 2163 | 18 | 6 | 0 | 14 |
| Bus | 21 | 2189 | 6 | 2 | 11 |
| Car | 0 | 1 | 2101 | 3 | 0 |
| Metro | 0 | 11 | 6 | 2139 | 2 |
| Walk | 45 | 5 | 8 | 3 | 2130 |

## 5. Conclusions

In this paper, we presented a low-cost and lightweight system for travel data collection, sensing and travel mode detection based on smartphone sensors. For collecting the different sensors data, we developed a prototype application. Compared with traditional classification model, we proposed a deep learning-based method that can automatically and accurately detect different travel mode. Five representative travel modes (walk, bike, bus, car, and metro) are detected using the proposed model. In the experimental analysis, we investigated the classification performance of the CNN under different hyperparameters and combinations of sensors. We also discussed the impact of different sensors combinations on classification accuracy for each type of travel mode. The experimental results show that the precision, recall and F value of the proposed method can reach more than 97%. In future research, we will focus on more other types of travel mode recognition methods, such as the currently popular travel modes such as shared electric vehicles.

**Author Contributions:** Conceptualization, L.G. and W.M.; methodology, L.S. and J.H.; software, L.Z. and W.Y.; validation, J.H. and J.P.; formal analysis, J.H.; investigation, J.H. and L.S.; resources, J.H. and W.M.; data curation, W.M.; writing—original draft preparation, W.M. and J.H.; writing—review and editing, L.G. and J.H.; visualization, L.G. and W.M.; supervision, L.G. and W.Y. All authors have read and agreed to the published version of the manuscript.

**Funding:** This research was funded by Project Funded by China Postdoctoral Science Foundation (2021TQ0369, 2019M653009), National Natural Science Foundation of China (42001324, 41701519 and 42001404, 62102456), Shenzhen Scientific Research and Development Funding Program (JCYJ20180305125058727, JCYJ20170412105839839), Nature Science Foundation of Shenzhen University (2019094), and Key R&D Program of Ningxia Autonomous Region: Ecological environment monitoring and platform development of ecological barrier protection system for Helan Mountain (2022CMG02014).

**Institutional Review Board Statement:** Not applicable.

**Informed Consent Statement:** Not applicable.

**Data Availability Statement:** Not applicable.

**Conflicts of Interest:** The authors declare no conflict of interest.

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
