# Peer review of "Convolutional Neural Network-Based Travel Mode Recognition Based on Multiple Smartphone Sensors"

_applsci, doi:10.3390/app12136511_

Round 1

Reviewer 1 Report

Overview

The authors propose a system to classify travel mode data collected from smartphones using trained convolutional neural networks. The system is smartphone based but this applies to the data collection. Training of the CNN must be done a PC or server. The system can be very useful and is of high social impact. The science is sound by the paper has a number of flaws that need to be addressed.

The results are encouraging but it would have been better to use publicly available datasets used by other authors in order to make like-with-like comparisions.

English Style

The paper contains many grammatical, syntactical, and orthographic errors and needs to be proof read properly. 

Some examples:

Line 33 - benficial and not benefit.

Line 37 - inefficient and not inefficiency.

Line 39 - detection and not detecting.

Line 42 - rapid and not repaid.

Line 56 - inefficient and not inefficiency.

There are many others.

Some other comments:

  1.  Why was the F-Measure evaluation metric used. This is a commonly used metric but there are many others.
  2. How much data was used? How many people were involved in data gathering? Was there any pre-processing?
  3. Why was there a 60%-40% train test split? Was a validation set used? How was hyper-parameter optimization performed without a validation set? The test set is, by common convention, only used once at the end of model evaluation.
  4. Were dropout layers considered?
  5. Were 2-D data formats, such as spectrograms, considered for the sensor data?
  6. Figure 5 too small and text barely readable.
  7. Many figures are too low resolution.

Summary:

This is interesting work that has a high social impact. The main problem is that the English style needs improvement and the paper needs a good proof read. Also, some methodology and design choices are not explained well or not explained at all.

Author Response

Dear Editor,

We sincerely appreciate the effect and time you’ve spent in reviewing our manuscript (ID: applsci-1512881). Indeed, the comments are thoughtful and helpful. Our response to the comments from the reviewers is listed in the following:

*******************************************************************************

Reviewer #1:

Comment 1 by Review #1: 

Overview:The paper contains many grammatical, syntactical, and orthographic errors and needs to be proof read properly. Some examples: Line 33 - benficial and not benefit. Line 37 - inefficient and not inefficiency. Line 39 - detection and not detecting. Line 42 - rapid and not repaid. Line 56 - inefficient and not inefficiency. There are many others.

∎ Response:  Thank you for your approval of the research topic. We have carefully checked and revised the language expression and spelling in the full text. Most grammar and spelling errors have been corrected. We have rewritten most of the original manuscript to make the revised manuscript more readable. In the revised manuscript, the introduction, related work, methods, and experimental results have been polished in language. We hope that the revised manuscript will meet the journal's publication requirements. Thank you for your valuable advice.

Comment 2 by Review #1:  Why was the F-Measure evaluation metric used. This is a commonly used metric but there are many others.

∎ Response:  Thank you for your comment. We appreciate your suggestion very much. In fact, our selection of evaluation metrics is based on different tasks. When we evaluate the effectiveness of a certain type of travel mode detection task, we use the F-value, and when we evaluate the results of all five travel modes recognition, the confusion matrix is used to verify the effectiveness of the proposed CNN method (section 4.4). Therefore, here, depending on the nature of the task, we adopt two evaluation metrics to verify the effectiveness of the proposed method.

Comment 3 by Review #1:   How much data was used? How many people were involved in data gathering? Was there any pre-processing?

∎ Response:  Thank you for the comment. The size of the dataset is shown in Table 2. For each travel mode, we collected more than 5000 data records. During the data collection process, we invited 5 participants to carry out data collection tasks in the entire Shenzhen area. These data are basically one-dimensional time series data, and the sampling frequency is 50Hz. During pre-processing, we segmented the data and normalized it. For details, see Section 3.3 and Section 3.4 (the modified part is shown in RED)

Comment 4 by Review #1:   Why was there a 60%-40% train test split? Was a validation set used? How was hyper-parameter optimization performed without a validation set? The test set is, by common convention, only used once at the end of model evaluation.

∎ Response:  Thank you for the valuable comment. In fact, during the experiment, we divided the data into two parts with a ratio of 6:4, the former is used for training the model, and the latter is divided into two parts with equal proportions, one of which is used to select hyperparameters , and the other part is used to test the effectiveness of the classification model in the real world. Therefore, the ratio of training set, validation set, and test set is 6:2:2 [1,2]. Sorry that we failed to illustrate this in our experiments. We have explained this in Section 4.1 of the revised manuscript, see the RED section of the revised manuscript.

[1] Géron, A. (2017). Hands-on machine learning with scikit-learn and tensorflow: Concepts. Tools, and Techniques to build intelligent systems.

[2] James, G., Witten, D., Hastie, T., & Tibshirani, R. (2013). An introduction to statistical learning (Vol. 112, p. 18). New York: springer.

Comment 5 by Review #1:   Were dropout layers considered?

∎ Response:  Thank you very much for your valuable comments. The purpose of the dropout layer is to prevent overfitting and is often designed for complex nonlinear classification tasks. However, during the experiment, we find that the characteristics of the data collected in this paper are relatively obvious, and the various types of sensor data are less abnormal in actual situations. Therefore, we can obtain better travel mode recognition results when we do not need to design the dropout layer. Therefore, no dropout layer is designed. Our goal is to provide a prototype application framework for travel mode recognition and to verify the effectiveness of the proposed method.

Comment 6 by Review #1:   Were 2-D data formats, such as spectrograms, considered for the sensor data?

∎ Response:  Thank you for the comment. We mainly use four kinds of sensor data to classify different travel mode, including accelerometer, gyroscope, magnetometer, and barometer. Since these data are recorded in time series, we mainly use one-dimensional data for training and classification, and spectrogram data is not included here. However, we believe that this is a type of data that is beneficial to improve classification accuracy, so we will focus on this type of data in future research.

Comment 7 by Review #1:   Figure 5 too small and text barely readable.

∎ Response:  Thank you for pointing out that. We improved the resolution and quality of Figure 5, the same of other figures. The text of Figure 5 is now more readable. Thank you for your advice.

Comment 8 by Review #1:   Many figures are too low resolution.

∎ Response:  Thank you for your advice. We revised all the related figures, and the resolution of all figures are checked to meet the requirement of this journal. Thanks for pointing out that.

Comment 9 by Review #1:  Summary: This is interesting work that has a high social impact. The main problem is that the English style needs improvement and the paper needs a good proof read. Also, some methodology and design choices are not explained well or not explained at all.

∎ Response:  Thank you VERY MUCH for pointing out that. I believe the value of this work, thus, to improve the quality of this paper, we refined the “Methods” and “Results” part, also, we described the methodology in detail in the revised part. Further, we have polished language expressions almost everywhere in the text to make this paper easier to read.

We thank your effort and time on this paper again. The modifications mentioned above are marked in RED in the revised manuscript. We hope that you and the reviewers will be satisfied with the revised manuscript.

Thanks, and best regards,

Yours Sincerely,

Dr. Jincai Huang

Big Data Institute, Central South University

Email: huangjincaicsu@csu.edu.cn, huangjincaicsu@gmail.com.

Reviewer 2 Report

The manuscript is like an experimental report. The architecture of the manuscript needs to be improved. It would be better if the authors introduced the methods before the experiments. The expression of the way should be separated from the experiment. Thus, it requires major revisions.

1) What are the authors' scientific contributions? Would you mind clarifying them?

2) The authors must refine the abstract section to indicate what the authors had done in 200 words.

3) In Sections 1 and 2, the authors must comment on the cited papers after introducing each relevant work. What readers require is, by convincing literature review, to understand the explicit thinking/consideration of why the proposed approach can reach more effective results. These are the very contribution from the authors.

4) The authors must introduce various recent research trends on CNN as follows:

- https://doi.org/10.3390/electronics10162004;

- https://doi.org/10.1007/s11042-018-5610-8;

- https://doi.org/10.3390/app10238476;

- https://doi.org/10.3837/tiis.2020.08.013.

In addition, the authors must provide a sufficient critical review of the literature to indicate the drawbacks of existing approaches and then define the main focus of the research direction. How did those previous studies perform? Specifically, what methodology did they use? Which problem still requires to be solved? Why is the proposed approach suitable for solving the critical problem? Readers need more positive reviews of the literature to indicate the state-of-the-art development.

5) In Section 3, the authors must introduce their proposed research framework more effectively. For example, the authors could consider some essential brief explanation compared to the text with a total research flowchart or framework diagram for each proposed algorithm to indicate how these employed models are working to receive the experimental results. It is not easy to understand how the proposed approaches work.

6) The authors must perform exploratory data analysis on the datasets.

7) In Section 4, the authors must use more alternative benchmarking models.

- https://doi.org/10.3390/s19204411.

8) The authors must also conduct some statistical tests to ensure the superiority of the proposed approach, i.e., how could the authors confirm that their results are superior to others? Meanwhile, the authors have also to provide an insightful discussion of the results. The authors can refer to the following references to conduct statistical tests.

- https://doi.org/10.3390/rs12091475

9) In Table 2, the authors must improve the table more precisely by providing the ranges of different parameters that they have considered for a grid search.

10) The authors must elevate the language of the article in general. Proofreading by multiple people familiar with the field of study will help in this regard.

Author Response

Dear Editor,

We sincerely appreciate the effect and time you’ve spent in reviewing our manuscript (ID: applsci-1512881). Indeed, the comments are thoughtful and helpful. Our response to the comments from the reviewers is listed in the following:

*******************************************************************************

Reviewer #2:

Comment 1 by Review #2:  The manuscript is like an experimental report. The architecture of the manuscript needs to be improved. It would be better if the authors introduced the methods before the experiments. The expression of the way should be separated from the experiment. Thus, it requires major revisions. 1)What are the authors' scientific contributions? Would you mind clarifying them?

∎ Response:  Thank you for pointing out that. We have reorganized the architecture of the whole paper, the "Methods" section details the CNN model used in this paper, and the "Results" section analyzes the experimental results. In the revised manuscript, the methods are described before the experiments and separated from each other. Our contributions to this article have been reorganized and stated in the “Introduction” section. The structure of the revised manuscript is clearer, and the content of each section does not overlap with each other. I hope readers will be satisfied with this version

Comment 2 by Review #2:  The authors must refine the abstract section to indicate what the authors had done in 200 words.

∎ Response:  Thank you for your comment. We had refined and revised the “Abstract” part, and the grammar and expression were carefully checked. We also kept this part within 200 words. Thanks again for your advice.

Comment 3 by Review #2:  In Sections 1 and 2, the authors must comment on the cited papers after introducing each relevant work. What readers require is, by convincing literature review, to understand the explicit thinking/consideration of why the proposed approach can reach more effective results. These are the very contribution from the authors.

∎ Response:  Thank you for pointing out that. We have reorganized the Section 1 and Section 2. We added some most recent papers to draw the main lines of current research. We reclassified related research by different source of data usage, as data availability remained one of the most important issue to detect different traffic modes, which would benefit the current traffic managements. Followed by this main line, we proposed a novel data collection and travel mode detection Android application to meet current challenges. With this main line, the literature review is now clearer. I hope the revised manuscript will meet the requirement for publication.

Comment 4 by Review #2:  The authors must introduce various recent research trends on CNN as follows: - https://doi.org/10.3390/electronics10162004;- https://doi.org/10.1007/s11042-018-5610-8;- https://doi.org/10.3390/app10238476; - https://doi.org/10.3837/tiis.2020.08.013.

In addition, the authors must provide a sufficient critical review of the literature to indicate the drawbacks of existing approaches and then define the main focus of the research direction. How did those previous studies perform? Specifically, what methodology did they use? Which problem still requires to be solved? Why is the proposed approach suitable for solving the critical problem? Readers need more positive reviews of the literature to indicate the state-of-the-art development.

∎ Response:  Thank you for pointing out that. In the revised manuscript, we reorganized the “Related Works” part. We added some most recent works, including the recommended papers by you, and moreover, we had sorted out the main lines of research in this field. We mainly organized the "Related Work" section of the revised manuscript according to the different data sources. Finally, we summarize the problems in the existing research and propose the research goals of this paper. Thank you for your comment again.

Comment 5 by Review #2:  In Section 3, the authors must introduce their proposed research framework more effectively. For example, the authors could consider some essential brief explanation compared to the text with a total research flowchart or framework diagram for each proposed algorithm to indicate how these employed models are working to receive the experimental results. It is not easy to understand how the proposed approaches work.

∎ Response:  Thank you for pointing out that. We re-described the proposed prototype system framework in this paper in the revised manuscript, and modified Figure 1 to illustrate the proposed prototype system architecture more effectively. We have explained the functions and structure of the client and server side, see section 3.1. Similarly, we also described the details of the CNN model. We hope this edition will meet the publication requirements.

Comment 6 by Review #2:  The authors must perform exploratory data analysis on the datasets.

∎ Response:  Thank you for the comment. The information on the data is described and summarized in section 4.1 of our revised manuscript. The size of the dataset is shown in Table 2. For each travel mode, we collected more than 5000 data records. During the data collection process, we invited 5 participants to carry out data collection tasks in the entire Shenzhen area. These data are basically one-dimensional time series data, and the sampling frequency is 50Hz. During pre-processing, we segmented the data and normalized it. (the modified part is shown in RED)

Comment 7 by Review #2:  In Section 4, the authors must use more alternative benchmarking models. - https://doi.org/10.3390/s19204411.

∎ Response:  Thank you for the comment. In this paper, we use four types of smartphone sensors, and the sensors used in the recent research are inconsistent with ours in number and type, so the format of the data we collect is different from that of existing studies, and comparative studies appear to exist some difficulties. Our main goal is to propose a prototype application to collect and classify travel modes, our focus is on the framework of the prototype system, so the data collection is real-time and realistic. In the experimental part, we analyze and verify the effect of different sensor data combinations. We are interested in the comparison between the different methods, whereas, we think that it is not the focus of this paper. In the future research, we will conduct an in-depth analysis of the comparison between the different methods. Now, according to your valuable comments, this paper has been greatly improved with careful revision, thank you again for your comments, and I hope this manuscript can meet the publication requirements.

Comment 8 by Review #2:  In Table 2, the authors must improve the table more precisely by providing the ranges of different parameters that they have considered for a grid search.

∎ Response:  Thank you for your comment. We have added another column of Table 2 to illustrate the range of these parameters. In fact, these parameters are relatively common in convolutional neural networks and have some conventional value ranges. To make the CNN model meet the needs of the task in this paper, we have introduced the range of parameter values, as shown in the revised manuscript.

Comment 9 by Review #2:  The authors must elevate the language of the article in general. Proofreading by multiple people familiar with the field of study will help in this regard.

∎ Response:  Thank you for pointing out that. We have made extensive revisions to the spelling and grammar of this manuscript, with careful consideration of almost every sentence to ensure correct expression. Especially, all figures in this manuscript have been enhanced to ensure the resolution required for publication. Moreover, we have reorganized the structure of the full text, so that the relevant work, methods, experiments, and other parts have a good degree of separation to ensure logical coherence and easy readability.

We thank your effort and time on this paper again. The modifications mentioned above are marked in RED in the revised manuscript. We hope that you and the reviewers will be satisfied with the revised manuscript.

Thanks, and best regards,

Yours Sincerely,

Dr. Jincai Huang

Big Data Institute, Central South University

Email: huangjincaicsu@csu.edu.cn, huangjincaicsu@gmail.com.

Reviewer 3 Report

The paper presents a solution to an interesting and valid problem, that can have many applications in real life. However, the topic itself is interesting the paper itself it not that engaging. In general there are many flaws in terms of merit and editing, but I think that when corrected the paper might present a sufficient level of merit and expertise for publication.

First of all, there are many unknowns in the solution design and testing that you should pay attention to. While reading the article, I had many key questions that were not answered in the text. Below I present a list of key questions and concerns in order in which they appear while reading the text of the paper:

  1. In the introduction, the authors state that their goal was to create a low-cost solution, but nowhere has it been defined what costs the authors hope to minimize. For this statement to be meaningful, it is necessary to introduce more qualitative measures than just the F-score and to compare the created solution with the previously published solutions. The lack of any comparison of the solution created with existing ones is one of the key problems in this paper.
  2. In its current form, the introduction is unfortunately very general and not very engaging, which in my opinion would have a negative impact on the readers' interest in this text. The paper does not explicitly describe the motivation to carry out presented research. In very general words, it is outlined that the detection of the mode of human movement can be used in transport management. However, there is no indication as to how the public or companies can benefit from such research results. In my opinion, pointing to specific examples instead of vague generalities would increase the interest of readers who are not experts in the field of neural networks. For example, indicating that the method of modeling urban transport allows to shorten the travel time by public transport, plan buses on the best routes or to reduce car traffic on the most loaded routes by introducing collective transport, which leads to reducing air pollution as well. Providing such examples together with specific percentage or numerical data showing the time or financial gain (I recommend looking at research on smart cities that show case studies in various cities around the world) would increase the readability and attractiveness of the text.
  3. The related work section is not complete in my opinion. There are no references to the latest research, including quite extensive comparative research, e.g. Paria Sadeghian, Johan Håkansson, Xiaoyun Zhao, Review and evaluation of methods in transport mode detection based on GPS tracking data,Journal of Traffic and Transportation Engineering (English Edition). Volume 8, Issue 4, 2021, Pages 467-482, ISSN 2095-7564.
  4. When describing solution in section 3.4 only ReLU activation function is mentioned. What made this particular function chosen, were others tested? I would like to know the process of making such valid design decisions. The same question applies to the optimization algorithm. 
  5. My most important remark concerns the solution and the testing phase descriptions, both covered in the Results section. First, there is no information regarding the implementation details and the test environment. Second, the lack of accurate information about the data used in the experiments is a very big oversight. You can guess that a new set of data has been created for the purposes of this research, however, I would require that it be directly indicated in the text and explained why the existing benchmark datasets were not used. It is important to indicate where the data came from, how many people have provided the data over what a period of time. In addition, I would also indicate in which geographic area these data were collected and whether they are publicly available anywhere.
  6. Regarding data set, another critical remark concerns how training and test datasets were created. The process of dividing the examined data set into these two subsets should be described. In the current version of the paper, it is not clear whether the training and testing data sets contained a similar percentage of data describing specific modes of movement/travel, so that respectively 60% and 40% of the data describing each specific travel mode were assigned to the training and test datasets, or whether the division was made randomly or using any other algorithm. I would expect a description of the method of data division taking into account the number of data samples describing individual modes of movement in the created training and testing data sets.
  7. Section 4.2 describes the effect of neural network parameters on one test metric, i.e. F-measure, and its components. First of all, I would like to know the reason why only this metric was chosen for optimization purposes and other measures of effectiveness/efficiency of the solution are not verified. In addition, individual experiments are not analyzed - there is a graph and 2-3 sentences of a comment that only summarize the results. I would expect a more detailed study of the results, e.g. by analyzing why the obtained results were better for some travel modes. In general, the research results should be analyzed, as at present most of them are more reported than described and explained.
  8. The paper lacks a comprehensive summary of the research carried out together with a comparison to the solutions previosly published. The existing research would constitute a point of reference indicating to what extent the created solution is innovative and whether it has a positive impact on the field of the problem under study.

Additionally, I have some minor comments, the introduction of which would certainly increase the readability and overall quality of the text:

  1. In the introduction, the authors state that initially the collection of data on human mobility was carried out as a result of direct and telephone surveys. The exact time frame when such studies formed the basis of data collection should be indicated. I believe that precision and accuracy when making such statements are crucial. In its present form, the said statement has no significant value, but only raises questions for beginners and very young researchers who might not know well the history of data collection and processing before the advent of mobile devices and the Internet.
  2. The paper requires a very thorough and professional editing of English language and style. Some sentences would require rephrasing to increase readability of the paper but some would require elimination of linguistic errors. Not being an expert in the English language itself, let me point out only the obvious examples.

a) Abstract: "So far, how to use these sensors data to accurate detect.." - to detect accurately

b) Abstract: "To collect the different sensors data, we developed a prototype application that not required setting in specific position or orientation while the application is running" - that does not require setting a smartphone in a specific....

c) Introduction:"putted" I believe a past tense of put is put

d) Related works "These methods only use GPS is limited in the area with sattelite Signal. But the places, like metro, are in shielded, where GPS cannot work." The quoted sentences need to be rewritten as they are not understandable. It seems to me that the authors meant that the methods using GPS are only valid in locations with access to satellite signal, and the metro, due to its underground location and the materials used in its construction, may not meet this condition.

e) In the section describing the related works, the sentences describing the individual works were initially written in the past tense, while at the end of the same section, they were written in the present tense. This is an inaccuracy and needs to be corrected - all sentences should be written consistently in the past tense.

3. The quality of the figures is too low, which is visible in print.

As part of this review, I focused on the aspects that require improvement, however, this research has been carried out to verify many aspects within an interesting and current problem area which might be of interest to readers. However, in my opinion it is not possible to accurately assess the merit of the paper and quality of the results without introducing the key changes that I have mentioned, especially comparison with existing solutions with indication what is special/novel in this research/solution.

Author Response

Dear Editor,

We sincerely appreciate the effect and time you’ve spent in reviewing our manuscript (ID: applsci-1512881). Indeed, the comments are thoughtful and helpful. Our response to the comments from the reviewers is listed in the following:

*******************************************************************************

Reviewer #3:

Comment 1 by Review #3:  The paper presents a solution to an interesting and valid problem, that can have many applications in real life. However, the topic itself is interesting the paper itself it not that engaging. In general, there are many flaws in terms of merit and editing, but I think that when corrected the paper might present a sufficient level of merit and expertise for publication.

First of all, there are many unknowns in the solution design and testing that you should pay attention to. While reading the article, I had many key questions that were not answered in the text. Below I present a list of key questions and concerns in order in which they appear while reading the text of the paper: In the introduction, the authors state that their goal was to create a low-cost solution, but nowhere has it been defined what costs the authors hope to minimize. For this statement to be meaningful, it is necessary to introduce more qualitative measures than just the F-score and to compare the created solution with the previously published solutions. The lack of any comparison of the solution created with existing ones is one of the key problems in this paper.

∎ Response:  Thank you for the approval of the proposed topic. We firstly made extensive revisions to the spelling and grammar of this manuscript to make this article more readable. Then, we have reorganized the structure of the full text to make it logically coherent. The modified part is shown in red in the main text.

In the “Introduction” part, we have recapitulated the main contributions of this paper, that is, to propose a prototype application to collect and classify travel modes, and to explore how to combine different sensors data to improve the precision. The data is able to collect by common smartphones, Since smartphones have become quite popular in China, thus, the solution is a low-cost way. We have recognized that it would be inappropriate to state this in the “Introduction” part, so we have re-polished the “Introduction” part in the revised manuscript. Thank you again for pointing out that.

Comment 2 by Review #3:  In its current form, the introduction is unfortunately very general and not very engaging, which in my opinion would have a negative impact on the readers' interest in this text. The paper does not explicitly describe the motivation to carry out presented research. In very general words, it is outlined that the detection of the mode of human movement can be used in transport management. However, there is no indication as to how the public or companies can benefit from such research results. In my opinion, pointing to specific examples instead of vague generalities would increase the interest of readers who are not experts in the field of neural networks. For example, indicating that the method of modeling urban transport allows to shorten the travel time by public transport, plan buses on the best routes or to reduce car traffic on the most loaded routes by introducing collective transport, which leads to reducing air pollution as well. Providing such examples together with specific percentage or numerical data showing the time or financial gain (I recommend looking at research on smart cities that show case studies in various cities around the world) would increase the readability and attractiveness of the text.

∎ Response:  Thank you for your comment. We have reorganized the “Introduction” part. To provide a more understandable way, we introduced concepts related to smart city and smart mobility, then, relevant examples are listed to illustrate the benefits of travel pattern recognition for smart mobility, including reducing travel time, reducing congestion, etc. We reviewed relevant smart city-related cases and re-expressed the introduction part to increase readers' interest in reading. Thank you again for your comment.

Comment 3 by Review #3:  The related work section is not complete in my opinion. There are no references to the latest research, including quite extensive comparative research, e.g. Paria Sadeghian, Johan Håkansson, Xiaoyun Zhao, Review and evaluation of methods in transport mode detection based on GPS tracking data, Journal of Traffic and Transportation Engineering (English Edition). Volume 8, Issue 4, 2021, Pages 467-482, ISSN 2095-7564.

∎ Response:  Thank you for the comment. To conduct a comprehensive review of the current research, we have added relevant research in recent years and sorted out the main lines of recent research. We have added citations to the papers you recommended to make the review of current research more comprehensive in the revised manuscript, thank you for your recommendation.

Comment 4 by Review #3:  When describing solution in section 3.4 only ReLU activation function is mentioned. What made this particular function chosen, were others tested? I would like to know the process of making such valid design decisions. The same question applies to the optimization algorithm.

∎ Response:  Thank you for your comment. We choose the ReLU activation function based on experience. In our previous programming experience, we believed that the gradient vanishing problem would not occur when using the ReLU function, that is, the ReLU function is linear when x>0 and has a fixed gradient. In addition, the ReLU activation function will output 0 (deactivation) when x<0, which can cause the sparsity of the network, which can well simulate the working principle of human brain neurons. What's more, the ReLU function also has the advantages of simple calculation. We considered BGD and SGD algorithms when choosing an optimization algorithm. The former has the disadvantage of slow training speed, and the latter may fall into the problem of local optimal solution. The Mini-Batch GD algorithm can greatly reduce the number of iterations required for convergence, and at the same time, it can make the converged result closer to the effect of gradient descent. We have tested these algorithms, in order to make the proposed CNN model optimal, we use the current ReLU activation function and Mini-Batch GD optimization algorithm.

Comment 5 by Review #3:  My most important remark concerns the solution and the testing phase descriptions, both covered in the Results section. First, there is no information regarding the implementation details and the test environment. Second, the lack of accurate information about the data used in the experiments is a very big oversight. You can guess that a new set of data has been created for the purposes of this research, however, I would require that it be directly indicated in the text and explained why the existing benchmark datasets were not used. It is important to indicate where the data came from, how many people have provided the data over what a period of time. In addition, I would also indicate in which geographic area these data were collected and whether they are publicly available anywhere.

∎ Response:  Thank you for pointing out that. In the revised manuscript, we first described the data size, format, and number of people collected in detail, see section 4.1 for details. In this paper, we use four types of smartphone sensors, and the sensors used in the recent research are inconsistent with ours in number and type, so the format of the data we collect is different from that of existing studies, and comparative studies appear to exist some difficulties. This paper’s focus is on the framework of the prototype system, so the data collection is real-time and realistic. The geographic area of these data collected here covers the whole Shenzhen city, and these data will be available on github (https://github.com/huangjincaicsu/Dataset) once this paper is published. In this way, peer scholars can easily use this data and conduct comparative analysis.

Comment 6 by Review #3:  Regarding data set, another critical remark concerns how training and test datasets were created. The process of dividing the examined data set into these two subsets should be described. In the current version of the paper, it is not clear whether the training and testing data sets contained a similar percentage of data describing specific modes of movement/travel, so that respectively 60% and 40% of the data describing each specific travel mode were assigned to the training and test datasets, or whether the division was made randomly or using any other algorithm. I would expect a description of the method of data division taking into account the number of data samples describing individual modes of movement in the created training and testing data sets.

∎ Response:  Thank you for pointing out that. In fact, during the experimental analysis, we divided the data into two parts with a ratio of 6:4, the former is used for training the model, and the latter is divided into two parts with equal proportions, one of which is used to select hyperparameters , and the other part is used to test the effectiveness of the classification model in the real world. Therefore, the ratio of training set, validation set, and test set is 6:2:2 [1,2]. Sorry that we failed to illustrate this in our experiments. We have explained this in Section 4.1 of the revised manuscript, see the RED section of the revised manuscript.

[1] Géron, A. (2017). Hands-on machine learning with scikit-learn and tensorflow: Concepts. Tools, and Techniques to build intelligent systems.

[2] James, G., Witten, D., Hastie, T., & Tibshirani, R. (2013). An introduction to statistical learning (Vol. 112, p. 18). New York: springer.

Comment 7 by Review #3: Section 4.2 describes the effect of neural network parameters on one test metric, i.e. F-measure, and its components. First of all, I would like to know the reason why only this metric was chosen for optimization purposes and other measures of effectiveness/efficiency of the solution are not verified. In addition, individual experiments are not analyzed - there is a graph and 2-3 sentences of a comment that only summarize the results. I would expect a more detailed study of the results, e.g. by analyzing why the obtained results were better for some travel modes. In general, the research results should be analyzed, as at present most of them are more reported than described and explained.

∎ Response:  Thank you for your comment. In classification tasks, precision or recall are generally used to evaluate the validity of experimental results, but they also have some shortcomings. For example, their denominators are different, resulting in relative evaluation results. In this paper, the F value is a trade-off between precision and recall, so it is adopted in this paper, which does not mean that precision and recall are invalid. In addition, the confusion matrix takes into account the misjudgment and missed judgment at the same time, which can evaluate the validity of the experimental results. Therefore, in the classification result statistics, we use the confusion matrix to evaluate. We have modified Section 4.2 to analyze some interpretable parameters in the experimental results, such as the number of layers and learning rate, and how they affect the experimental results. However, other parameters, such as the number of feature maps and batch size, have ambiguous effects on the experimental results. For example, batch size is the amount of data for each training. It is difficult for us to find a specific logical connection, so we only analyze experimental phenomena to choose the best parameters.

Comment 8 by Review #3:  The paper lacks a comprehensive summary of the research carried out together with a comparison to the solutions previously published. The existing research would constitute a point of reference indicating to what extent the created solution is innovative and whether it has a positive impact on the field of the problem under study.

∎ Response:  Thank you for pointing out that. We believe it is important for every paper to review and summarize existing literature and explain to what extent the proposed method is innovative. To achieve this, we have reorganized the introductory part, showing the significance of travel pattern recognition for smart cities, and teasing out the context of existing methods, and explained the insufficiency of existing research. On this basis, we proposed the research ideas of this paper, and point out the contributions of the proposed method. Secondly, for the summary of the existing research, we carried out a detailed description in the related work section, and in the last paragraph, we summarize the existing research work. In this way, we try to make the innovation of this paper clearly visible. Thank you again.

Minor revisions:

Comment 9 by Review #3:  Additionally, I have some minor comments, the introduction of which would certainly increase the readability and overall quality of the text: In the introduction, the authors state that initially the collection of data on human mobility was carried out as a result of direct and telephone surveys. The exact time frame when such studies formed the basis of data collection should be indicated. I believe that precision and accuracy when making such statements are crucial. In its present form, the said statement has no significant value, but only raises questions for beginners and very young researchers who might not know well the history of data collection and processing before the advent of mobile devices and the Internet.

∎ Response:  Thank you for pointing out that. We explored the history of household travel surveys, we found that the study was almost 40 years old [1], so we added the time to the traditional method of travel mode recognition (home travel surveys) in the introduction section, and for the age of the data source (GPS data) used is presented. In this way, we hope to help very young researchers understand the detail of related research who might not know well the history of data collection.

[1] Wittwer, R., Hubrich, S., Wittig, S., & Gerike, R. (2018). Development of a new method for household travel survey data harmonisation. Transportation research procedia, 32, 597-606.

Comment 10 by Review #3:  The paper requires a very thorough and professional editing of English language and style. Some sentences would require rephrasing to increase readability of the paper but some would require elimination of linguistic errors. Not being an expert in the English language itself, let me point out only the obvious examples.

  1. a) Abstract: "So far, how to use these sensors data to accurate detect.." - to detect accurately
  2. b) Abstract: "To collect the different sensors data, we developed a prototype application that not required setting in specific position or orientation while the application is running" - that does not require setting a smartphone in a specific....
  3. c) Introduction:"putted" I believe a past tense of put is put
  4. d) Related works "These methods only use GPS is limited in the area with sattelite Signal. But the places, like metro, are in shielded, where GPS cannot work." The quoted sentences need to be rewritten as they are not understandable. It seems to me that the authors meant that the methods using GPS are only valid in locations with access to satellite signal, and the metro, due to its underground location and the materials used in its construction, may not meet this condition.
  5. e) In the section describing the related works, the sentences describing the individual works were initially written in the past tense, while at the end of the same section, they were written in the present tense. This is an inaccuracy and needs to be corrected - all sentences should be written consistently in the past tense.

∎ Response:  Thank you for your comment. Based on your comment, the grammatical errors you mentioned above have been corrected by us one by one, Secondly, we rewrote the abstract and reorganized its logical structure. We then polished and rewrote the introduction and related work section. To further improve the quality of this article, we have carefully checked all grammar and spelling throughout the full text to ensure that the expressions are accurate. All revised places are marked in RED, and it can be seen that most of the sentences in the revised manuscript have been carefully polished. Thank you for your valuable comment, I hope the revised manuscript will meet the publication requirements.

Comment 11 by Review #3:  The quality of the figures is too low, which is visible in print. As part of this review, I focused on the aspects that require improvement, however, this research has been carried out to verify many aspects within an interesting and current problem area which might be of interest to readers. However, in my opinion it is not possible to accurately assess the merit of the paper and quality of the results without introducing the key changes that I have mentioned, especially comparison with existing solutions with indication what is special/novel in this research/solution.

∎ Response:  Thank you very much for your valuable comments. In fact, these comments are of great help in improving the quality of this paper. In response to your comments above, we have made the following changes: 1) We have made extensive revisions to the spelling and grammar of this manuscript, with careful consideration of almost every sentence to ensure correct expression. 2) All figures in this manuscript have been enhanced to ensure the resolution required for publication. 3) We have reorganized the structure of the full text, so that the relevant work, methods, experiments, and other parts have a good degree of separation to ensure logical coherence and easy readability. 4) Our contributions to this article have been condensed to clarify how this paper differs from others. 5) In this paper, we use four types of smartphone sensors, and the sensors used in the recent research are inconsistent with ours in number and type, so the format of the data we collect is different from that of existing studies, and comparative studies appear to exist some difficulties. Our main goal is to propose a prototype application to collect and classify travel modes, our focus is on the framework of the prototype system, so the data collection is real-time and realistic. In the experimental part, we analyze and verify the effect of different sensor data combinations. We are interested in the comparison between the different methods, whereas, we think that it is not the focus of this paper. In the future research, we will conduct an in-depth analysis of the comparison between the different methods. Now, according to your valuable comments, this paper has been greatly improved with careful revision, thank you again for your comments, and I hope this manuscript can meet the publication requirements.

We thank your effort and time on this paper again. The modifications mentioned above are marked in RED in the revised manuscript. We hope that you and the reviewers will be satisfied with the revised manuscript.

Thanks, and best regards,

Yours Sincerely,

Dr. Jincai Huang

Big Data Institute, Central South University

Email: huangjincaicsu@csu.edu.cn, huangjincaicsu@gmail.com.

Round 2

Reviewer 2 Report

The author addresses all the questions raised in the first round in the current version of the manuscript. 

This document is suitable for publication in Applied Sciences.